# Improving Gradient Estimation in Evolutionary Strategies With Past Descent Directions

## Abstract

Evolutionary Strategies (ES) are known to be an effective black-box optimization technique for deep neural networks when the true gradients cannot be computed, such as in Reinforcement Learning. We continue a recent line of research that uses surrogate gradients to improve the gradient estimation of ES. We propose a novel method to optimally incorporate surrogate gradient information. Our approach, unlike previous work, needs no information about the quality of the surrogate gradients and is always guaranteed to find a descent direction that is better than the surrogate gradient. This allows to iteratively use the previous gradient estimate as surrogate gradient for the current search point. We theoretically prove that this yields fast convergence to the true gradient for linear functions and show under simplifying assumptions that it significantly improves gradient estimates for general functions. Finally, we evaluate our approach empirically on MNIST and reinforcement learning tasks and show that it considerably improves the gradient estimation of ES at no extra computational cost.

## 1 Introduction

Evolutionary Strategies (ES) (1; 2; 3) are a black-box optimization technique, that estimate the gradient of some objective function with respect to the parameters by evaluating parameter perturbations in random directions. The benefits of using ES in Reinforcement Learning (RL) were exhibited in (4). ES approaches are highly parallelizable and account for robust learning, while having decent data-efficiency. Moreover, black-box optimization techniques like ES do not require propagation of gradients, are tolerant to long time horizons, and do not suffer from sparse reward distributions (4). This lead to a successful application of ES in variety of different RL settings (5; 6; 7; 8). Applications of ES outside RL include for example meta learning (9).

In many scenarios, the true gradient is impossible to compute, however surrogate gradients are available. Here, we use the term *surrogate gradients* for directions that are correlated but usually not equal to the true gradient, e.g. they might be biased or unbiased approximations of the gradient. Such scenarios include models with discrete stochastic variables (10), learned models in RL like Q-learning (11), truncated backpropagation through time (12) and feedback alignment (13), see (14) for a detailed exhibition. If surrogate gradients are available, it is beneficial to preferentially sample parameter perturbations from the subspace defined by these directions (14). The proposed algorithm (14) requires knowing in advance the quality of the surrogate gradient, does not always provide a descent direction that is better than the surrogate gradient, and it remains open how to obtain such surrogate gradients in general settings.

In deep learning in general, experimental evidence has established that higher order derivatives are usually "well behaved", in which case gradients of consecutive parameter updates correlate and applying momentum speeds up convergence (15; 16; 17). These observations suggest that past update directions are promising candidates for surrogate gradients.

In this work, we extend the line of research of (14). Our contribution is threefold:

- First, we show theoretically how to optimally combine the surrogate gradient directions with random search directions. More precisely, our approach computes the direction of the subspace spanned by the evaluated search directions that is most aligned with the true gradient. Our gradient estimator does not need to know the quality of the surrogate gradients

and always provides a descent direction that is more aligned with the true gradient than the surrogate gradient.

- Second, above properties of our gradient estimator allow us to iteratively use the last update direction as a surrogate gradient for our gradient estimator. Repeatedly using the last update direction as a surrogate gradient will aggregate information about the gradient over time and results in improved gradient estimates. In order to demonstrate how the gradient estimate improves over time, we prove fast convergence to the true gradient for linear functions and show, that under simplifying assumptions, it offers an improvement over ES that depends on the Hessian for general functions.

- Third, we validate experimentally that these results transfer to practice, that is, the proposed approach computes more accurate gradients than standard ES. We observe that our algorithm considerably improves gradient estimation on the MNIST task compared to standard ES and that it improves convergence speed and performance on the tested Roboschool reinforcement learning environments.

## 2 RELATED WORK

Evolutionary strategies (1; 2; 3) are black box optimization techniques that approximate the gradient by sampling finite differences in random directions in parameter space. Promising potential of ES for the optimization of neural networks used for RL was demonstrated in (4). They showed that ES gives rise to efficient training despite the noisy gradient estimates that are generated from a much smaller number of samples than the dimensionality of parameter space. This placed ES on a prominent spot in the RL tool kit (5; 6; 7; 8).

The history of descent directions was previously used to adapt the search distribution in covariance matrix adaptation ES (CMA-ES) (18). CMA-ES constructs a second-order model of the underlying objective function and samples search directions and adapts step size according to it. However, maintaining the full covariance matrix makes the algorithm quadratic in the number of parameters, and thus impractical for high-dimensional spaces. Linear time approximations of CMA-ES like diagonal approximations of the covariance matrix (19) often do not work well, in the sense that their gradient estimates do not converge to the true gradient even if the true gradient does not change over time. Our approach differs as we simply improve the gradient estimation and then feed the gradient estimate to a first-order optimization algorithm.

Our work is inspired by the line of research of (14), where surrogate gradient directions are used to improve gradient estimations by 'elongating' the search space along these directions. That approach has two shortcomings. First, the bias of the surrogate gradients needs to be known to adapt the covariance matrix. Second, once the bias of the surrogate gradient is too small, the algorithm will not find a better descent direction than the surrogate gradient.

Another related area of research investigates how to use momentum for the optimization of deep neural networks. Applying different kinds of momentum has become one of the standard tools in current deep learning and it has been shown to speed-up learning in a very wide range of tasks (20; 16; 17). This hints, that for many problems the higher-order terms in deep learning models are "well-behaved" and thus, the gradients do not change too drastically after parameter updates. While these approaches use momentum for parameter updates, our approach can be seen as a form of momentum when sampling directions from the search space of ES.

## 3 GRADIENT ESTIMATION

We aim at minimizing a function $f : \mathbb{R}^n \to \mathbb{R}$ by steepest descent. In scenarios where the gradient $\bigtriangledown f$ does not exist or is inefficient to compute, we are interested in obtaining some estimate of the (smoothed) gradient of $f$ that provides a good parameter update direction.

### 3.1 THE ES GRADIENT ESTIMATOR

ES considers the function $f_\sigma$ that is obtained by *Gaussian smoothing*

$$f_\sigma(\theta) = \mathbb{E}_{\epsilon \sim \mathcal{N}(\mathbf{0}, \mathrm{I})}[f(\theta + \sigma\epsilon)] \, ,$$

where $\sigma$ is a parameter modulating the size of the smoothing area and $\mathcal{N}(\mathbf{0}, \mathrm{I})$ is the $n$-dimensional Gaussian distribution with $\mathbf{0}$ being the all $0$ vector and I being the $n$-dimensional identity matrix. The gradient of $f_\sigma$ with respect to parameters $\theta$ is given by

$$\bigtriangledown f_\sigma = \frac{1}{\sigma} \mathbb{E}_{\epsilon \sim \mathcal{N}(\mathbf{0}, \mathrm{I})}[f(\theta + \sigma\epsilon)\epsilon],$$

which can be sampled by a Monte Carlo estimator, see (5). Often antithetic sampling is used, as it reduces variance (5). The *antithetic ES gradient estimator* using $P$ samples is given by

$$g_{ES} = \frac{1}{2\sigma P} \sum_{i=1}^{P} \left( f(\theta + \sigma\epsilon^i) - f(\theta - \sigma\epsilon^i) \right) \epsilon^i , \tag{1}$$

where $\epsilon^i$ are independently sampled from $\mathcal{N}(\mathbf{0}, \mathrm{I})$ for $i \in \{1, \ldots, P\}$. This gradient estimator has been shown to be effective in RL settings (4).

## 3.2 OUR ONE STEP GRADIENT ESTIMATOR

We first give some intuition before presenting our gradient estimator formally. Given one surrogate gradient direction $\zeta$, our one step gradient estimator applies the following sampling strategy. First, it estimates how much the gradient points into the direction of $\zeta$ by antithetically evaluating $f$ in the direction of $\zeta$. Second, it estimates the part of $\bigtriangledown f$ that is orthogonal to $\zeta$ by evaluating random, pairwise orthogonal search directions that are orthogonal to $\zeta$. In this way, our estimator detects the optimal lengths of the parameter update step into both the surrogate direction and the evaluated orthogonal directions (e.g. if $\zeta$ and $\bigtriangledown f$ are parallel, the update step is parallel to $\zeta$, and if they are orthogonal the step into direction $\zeta$ has length $0$). Additionally, if the surrogate direction and the gradient are not perfectly aligned, then the gradient estimate almost surely improves over the surrogate direction due to the contribution from the evaluated directions orthogonal to $\zeta$. In the following we define our estimator formally and prove that the estimated direction possesses best possible alignment with the gradient that can be achieved with our sampling scheme.

We assume that $k$ pairwise orthogonal surrogate gradient directions $\zeta^1, \ldots, \zeta^k$ are given to our estimator. Denote by $\mathbb{R}_\zeta$ the subspace of $\mathbb{R}^n$ that is spanned by the $\zeta^i$, and by $\mathbb{R}_{\perp\zeta}$ the subspace that is orthogonal to $\mathbb{R}_\zeta$. Further, for vectors $v$ and $\bigtriangledown f$, we denote by $\hat{v}$ and $\hat{\bigtriangledown} f$ the normalized vector $\frac{v}{\|v\|}$ and $\frac{\bigtriangledown f}{\|\bigtriangledown f\|}$, respectively. Let $\hat{\epsilon}^1, \ldots, \hat{\epsilon}^P$ be random orthogonal unit vectors from $\mathbb{R}_{\perp\zeta}$. Then, our estimator is defined as

$$g_\perp = \sum_{i=1}^{k} \frac{f(\theta + \sigma\hat{\zeta}^i) - f(\theta - \sigma\hat{\zeta}^i)}{2\sigma} \hat{\zeta}^i + \sum_{i=1}^{P} \frac{f(\theta + \sigma\hat{\epsilon}^i) - f(\theta - \sigma\hat{\epsilon}^i)}{2\sigma} \hat{\epsilon}^i . \tag{2}$$

We write $\bigtriangledown f = \bigtriangledown f_{\|\zeta} + \bigtriangledown f_{\perp\zeta}$, where $\bigtriangledown f_{\|\zeta}$ and $\bigtriangledown f_{\perp\zeta}$ are the projections of $\bigtriangledown f$ on $\mathbb{R}_\zeta$ and $\mathbb{R}_{\perp\zeta}$, respectively. In essence, the first sum in (2) computes $\bigtriangledown f_{\|\zeta}$ by assessing the quality of each surrogate gradient direction, and the second sum estimates $\bigtriangledown f_{\perp\zeta}$ similar to an orthogonalized antithetic ES gradient estimator, that samples directions from $\mathbb{R}_{\perp\zeta}$, see (5). We remark that we require pairwise orthogonal unit directions $\hat{\epsilon}^i$ for the optimality proof. Due to the orthogonality of the directions, no normalization factor like the $1/P$ factor in (1) is required in (2). In practice, the dimensionality $n$ is often much larger than $P$. Then, sampling pairwise orthogonal unit vecotrs $\epsilon^i$ is nearly identical to sampling the $\epsilon^i$s from a $\mathcal{N}(\mathbf{0}, \mathrm{I})$ distribution, because in high-dimensional space the norm of $\epsilon^i \sim \mathcal{N}(\mathbf{0}, \mathrm{I})$ is highly concentrated around $1$ and the cosine of two such random vectors is highly concentrated around $0$.

For the sake of analysis, we assume that $f$ is differentiable and that the second order approximation $f(x + \epsilon) \approx f(x) + \langle \epsilon, \bigtriangledown f(x) \rangle + \epsilon^T H(x)\epsilon$, where $H(x)$ denotes the Hessian matrix of $f$ at $x$, is exact. This assumption implies that

$$\frac{f(\theta + \sigma\hat{\epsilon}) - f(\theta - \sigma\hat{\epsilon})}{2\sigma} = \langle \bigtriangledown f(\theta), \hat{\epsilon} \rangle , \tag{3}$$

because the even terms cancel for antithetic sampling. The following proposition and theorems provide theoretical understanding, how our gradient estimation scheme improves gradient estimation

in this smooth, noise-free setting. In the following, we will omit the $\theta$ in $\triangledown f(\theta)$. Our first proposition states that $g_\perp$ computes the direction in the subspace spanned by $\zeta^1, \ldots, \zeta^k, \epsilon^1, \ldots, \epsilon^P$ that is most aligned with $\triangledown f$.

**Proposition 1** (Optimality of $g_\perp$)**.** *Let* $\zeta^1, \ldots, \zeta^k, \epsilon^1, \ldots, \epsilon^P$ *be pairwise orthogonal vectors in* $\mathbb{R}^n$. *Then,* $g_\perp = \sum_{i=1}^{k} \langle \triangledown f, \hat{\zeta}^i \rangle \hat{\zeta}^i + \sum_{i=1}^{P} \langle \triangledown f, \hat{\epsilon}^i \rangle \hat{\epsilon}^i$ *computes the projection of* $\triangledown f$ *on the subspace spanned by* $\zeta^1, \ldots, \zeta^k, \epsilon^1, \ldots, \epsilon^P$. *Especially,* $\epsilon = g_\perp$ *is the vector of that subspace that maximizes the cosine* $\langle \hat{\triangledown} f, \hat{\epsilon} \rangle$ *between* $\triangledown f$ *and* $\epsilon$. *Moreover, the squared cosine between* $g_\perp$ *and* $\triangledown f$ *is given by*

$$\langle \hat{\triangledown} f, \hat{g}_{our} \rangle^2 = \sum_{i=1}^{k} \langle \hat{\triangledown} f, \hat{\zeta}^i \rangle^2 + \sum_{i=1}^{P} \langle \hat{\triangledown} f, \hat{\epsilon}^i \rangle^2 . \tag{4}$$

We remark that when evaluating $\langle \triangledown f, v^i \rangle$ for arbitrary directions $v^i$, no information about search directions orthogonal to the subspace spanned by the $v^i$s is obtained. Therefore, one can only hope for finding the best approximation of $\triangledown f$ lying within the subspace spanned by the $v^i$s, which is accomplished by $g_\perp$. The proof of Proposition 1 follows easily from the Cauchy-Schwarz inequality and is given in the appendix.

### 3.3 ITERATIVE GRADIENT ESTIMATION USING PAST DESCENT DIRECTIONS

Our gradient estimation algorithm iteratively applies the one step gradient estimator $g_\perp$ by using the gradient estimate of the last time step as surrogate direction for the current time step. Therefore, our algorithm relies on the assumption that gradients are correlated from one time step to the next. This assumption is justified since it is one of the reasons momentum-based optimizers (15; 16; 17) are successful in deep learning. We also explicitly test this assumption experimentally, see Figure 1a. The quality of the gradient estimate at any time step, depends on the quality of the surrogate gradient, which might be restrained because (a) the previous gradient estimate might be very noisy, and (b) the gradient changes with parameter updates. Our algorithm efficiently tackles problem (a), since it improves the gradient estimate over the surrogate gradient at any time step. In order to theoretically quantify this, we first analyse the case where the gradient does not change with parameter updates, i.e. if a linear function is optimized. In this setting, we show that our algorithm needs a small factor more than $n$ (dimension) samples to align with the true gradient, see Theorem 1, which is close to optimal, because at least $n$ samples are required to determine the true gradient. Next, we incorporate problem (b) into our analysis by considering general, non-linear functions, for which the gradient changes with parameter updates. We show under some simplifying assumptions that, also in this case, our algorithm builds up an improved gradient estimate over time, see Theorem 2.

We first need some notation. Denote by $\theta_t$ the search point, by $\triangledown f_t = \triangledown f(\theta_t)$ the gradient and by $\zeta_t$ the parameter update step at time $t$, that is, $\theta_{t+1} = \theta_t + \zeta_t$. The iterative gradient estimation algorithm obtains the gradient estimate $\zeta_t$ by computing $g_\perp$ with the last update direction $\zeta_{t-1}$ as surrogate gradient and $P$ new random directions $\hat{\epsilon}^i$. Formally, let $\hat{\epsilon}_t^1, \ldots, \hat{\epsilon}_t^P$ be pairwise orthogonal unit directions chosen uniformly from the unit sphere, that is, they are conditioned to be pairwise orthogonal and are marginally uniformly distributed. By defining $\epsilon_t = \sum_{i=1}^{P} \langle \triangledown f_t, \hat{\epsilon}_t^i \rangle \hat{\epsilon}_t^i$, and setting $\zeta_t = g_\perp$, we obtain

$$\zeta_t = \langle \triangledown f_t, \hat{\zeta}_{t-1} \rangle \hat{\zeta}_{t-1} + \sum_{i=1}^{P} \langle \triangledown f_t, \hat{\epsilon}_t^i \rangle \hat{\epsilon}_t^i = \langle \triangledown f_t, \hat{\zeta}_{t-1} \rangle \hat{\zeta}_{t-1} + \langle \triangledown f_t, \hat{\epsilon}_t \rangle \hat{\epsilon}_t , \tag{5}$$

where we used $\epsilon_t = \|\epsilon_t\|^2 / \|\epsilon_t\| \cdot \hat{\epsilon}_t = \langle \triangledown f_t, \epsilon_t \rangle / \|\epsilon_t\| \cdot \hat{\epsilon}_t = \langle \triangledown f, \hat{\epsilon}_t \rangle \hat{\epsilon}_t$. Then, Equation 4 of Proposition 1 turns into

$$\langle \hat{\triangledown} f_t, \zeta_t \rangle^2 = \langle \hat{\triangledown} f_t, \hat{\zeta}_{t-1} \rangle^2 + \langle \hat{\triangledown} f_t, \hat{\epsilon}_t \rangle^2 . \tag{6}$$

The next theorem quantifies how fast the cosine between $\zeta_t$ and $\triangledown f_t$ converges to 1, if $\triangledown f_t$ does not change over time.

**Theorem 1** (Convergence rate for linear functions)**.** *Let* $\zeta_t$ *be iteratively computed, as in Equation* (5), *using the past update direction and* $P$ *pairwise orthogonal random directions and let* $X_t = \langle \hat{\triangledown} f, \hat{\zeta}_t \rangle$ *be the random variable that denotes the cosine between* $\zeta_t$ *and* $\triangledown f_t$ *at time* $t$. *Then, the expected*

*drift of $X_t^2$ is $\mathbb{E}[X_t^2 - X_{t-1}^2 | X_{t-1} = x_{t-1}] = (1 - x_{t-1}^2)\frac{P}{N-1}$. Moreover, let $\epsilon > 0$ and define $T$ to be the first point in time $t$ with $X_t^2 \geq 1 - \delta$. It holds*

$$\mathbb{E}[T] \leq \frac{N-1}{P} \min\{(1-\delta)/\delta, 1 + \ln(1/\delta)\}\,.$$

The first bound $\mathbb{E}[T] \leq \frac{N-1}{P}\frac{1-\delta}{\delta}$ is tight for $\delta$ close to 1 and follows by an additive drift theorem, while the second bound $\mathbb{E}[T] \leq \frac{N-1}{P}(1 + \ln(1/\delta))$ is tight for $\delta$ close to 0 and follows by a variable drift theorem, see appendix. We remark that in a smooth and noise-free setting, where one can sample the true directional derivative with Equation (3), a cosine squared of $1 - \delta$ can be reached by sampling $(1 - \delta)N$ random orthogonal directions, see Proposition 2 in the appendix. Since our algorithm evaluates $P + 1$ directions per time step, it requires approximately $\min\{1/\delta, \frac{1+\ln(1/\delta)}{1-\delta}\}$ times more samples to reach the same alignment.

Naturally, the linear case is not the most interesting one. However, it is hard to rigorously analyse the case of general $f$, because it is unpredictable how the gradient $\bigtriangledown f_t$ differs from $\bigtriangledown f_{t-1}$. Note that $\bigtriangledown f_t - \bigtriangledown f_{t-1} \approx H\zeta_{t-1}$, where $H$ is the Hessian matrix of $f$ at $\theta_{t-1}$. We define $\alpha_t = \langle \hat{\bigtriangledown} f_t, \hat{\bigtriangledown} f_{t-1}\rangle$ and write $\hat{\bigtriangledown} f_t = \alpha_t \hat{\bigtriangledown} f_{t-1} + \bigtriangledown f_\perp$ where $\bigtriangledown f_\perp$ is orthogonal to $\hat{\bigtriangledown} f_t$ and has squared norm $1 - \alpha_t^2$. Then, the first term of (6) is equal to

$$\langle \hat{\bigtriangledown} f_t, \hat{\zeta}_{t-1}\rangle^2 = \langle \alpha_t \hat{\bigtriangledown} f_{t-1} + \bigtriangledown f_\perp, \hat{\zeta}_{t-1}\rangle^2 = \left(\alpha_t \langle \hat{\bigtriangledown} f_{t-1}, \hat{\zeta}_{t-1}\rangle + \langle \bigtriangledown f_\perp, \hat{\zeta}_{t-1}\rangle\right)^2\,.$$

In the following, we assume that $\bigtriangledown f_\perp$ is a direction orthogonal to $\bigtriangledown f_{t-1}$ chosen uniformly at random. Though, this assumption is not entirely true, it allows to get a grasp on the approximate cosine that our estimator is going to converge to.

**Theorem 2.** *Let $\zeta_t$ be iteratively computed using the past update direction and $P$ pairwise orthogonal random directions, see Equation (5), and let $X_t = \langle \hat{\bigtriangledown} f, \hat{\zeta}_t\rangle$ be the random variable that denotes the cosine between $\zeta_t$ and $\bigtriangledown f_t$ at time $t$. Further, write $\hat{\bigtriangledown} f_t = \alpha_t \hat{\bigtriangledown} f_{t-1} + \bigtriangledown f_\perp$, where $1 \geq \alpha_t \geq 0$ and assume that $\bigtriangledown f_\perp$ has a random direction orthogonal to $\hat{\bigtriangledown} f_{t-1}$. Choose $\zeta_t$ according to Equation (5) and define $X_t$ to be the cosine between $\hat{\bigtriangledown} f_t$ and $\hat{\zeta}_t$. Then,*

$$\mathbb{E}[X_t^2 | X_{t-1} = x_{t-1}] = \left(\alpha_t^2 x_{t-1}^2 + (1 - \alpha_t^2)(1 - x_{t-1}^2)\frac{1}{N-1}\right)\left(1 - \frac{P}{N-1}\right) + \frac{P}{N-1}\,.$$

The last theorem implies that the evolution of the cosine depends heavily on the cosine $\alpha_t$ between consecutive gradients. Let $A = \frac{(1-\alpha_t^2)\frac{1}{N-1}(1-\frac{P}{N-1})+\frac{P}{N-1}}{1-(\alpha_t^2+(1-\alpha_t^2)\frac{1}{N-1})(1-\frac{P}{N-1})}$. Then, the theorem implies that the drift $\mathbb{E}[X_t^2 - X_{t-1}^2 | X_{t-1} = x_{t-1}]$ is positive if $x_{t-1} \leq A$ and negative otherwise. Thus, if $\alpha_t$ would not change over time, we would expect $X_t$ to converge to $A$.

**TODO:**explain comparison to practise plot: The predicted alignement of the gradient estimate with the true gradient of Theorem 2 is plotted in Figure 1b. Though the assumptions of Theorem 2 are technically not true, the close fit of the theoretical prediction gives empirical evidence that the analysis captures the general behaviour of the gradient estimation process.

## 4 Experiments

In this section, we will empirically evaluate the performance of our gradient estimation scheme when combined with deep neural networks. In Section 4.1, we show that it significantly improves gradient estimation for digit classifiers on MNIST. In Section 4.2, we suggest how to overcome issues that arise from function evaluation noise. Finally, in Section 4.3, we evaluate our gradient estimation scheme on RL environments and investigate further issues arising in this setting.

### 4.1 Gradient Estimation and Performance on MNIST

We observe that our approach significantly improves gradient estimation compared to standard ES. Figure 1a shows that the key requisite of our iterative gradient estimation scheme is satisfied during

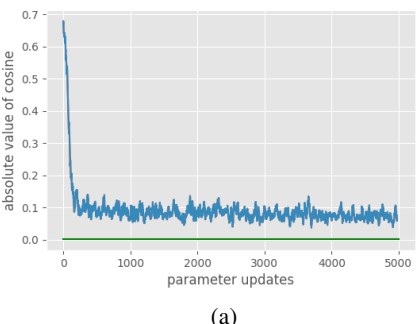 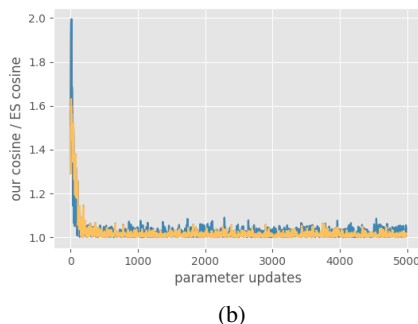

(a)  (b)

Figure 1: Improved gradient estimation. (a) The gradients before and after a parameter update are highly correlated. The cosine between two consecutive gradients (blue line) and the cosine between two random vectors (green line) are plotted. (b) A network is trained with parameter updates according to $g_{ES}$ using SGD (blue line) and Adam (yellow line) as optimizers. At any step we compute our gradient estimate with $g_{our}$ and the true gradient $\triangledown f$ with backpropagation. The plot shows that the ratio of the cosine between $g_{our}$ and $\triangledown f$ and the cosine between $g_{ES}$ and $\triangledown f$ is always strictly larger than 1.

training on MNIST, that is, that gradients between consecutive parameter update steps are correlated. Figure 1b shows that our approach improves gradient estimation compared to ES during the whole training process and strongly improves it in the beginning of training, where consecutive gradients are most correlated, see Figure 1a. We observe that our approach strongly outperforms ES in convergence speed and reaches better final performance for all hyperparameters we tested, see Figure 2 and Table 1.

**Implementation details:** For these experiments, we used a fully connected neural network with two hidden layers with a $tanh$ non-linearity and 1000 units each, to have a high dimensional model ($\sim 1.8$ million parameters) . For standard ES 128 random search directions are evaluated at each step. For our algorithm the previous gradient estimate and 126 random search directions are evaluated. We evaluated all directions on the same batch of images in order eliminate function evaluation noise and we resampled after every update step. We used small parameter perturbations ($\sigma = 0.001$). This is possible because no function evaluation noise is present and because the objective function is already differentiable and therefore no smoothing is required. We test both SGD and Adam optimizers with learning rates in the range $10^{0.5}, 10^0, \ldots, 10^{-3}$.

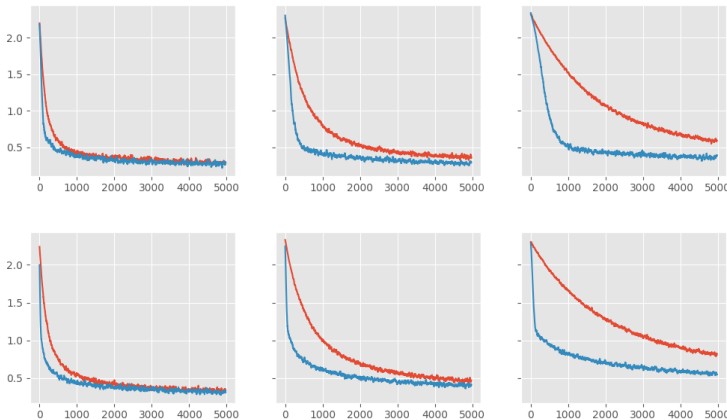

Figure 2: Performance of ES (red lines) and our algorithm (blue lines) on MNIST classification. The evolution of the training log-likelihood is plotted for the best three learning rates found for ES when using the Adam optimizer (top) and SGD (bottom). Our algorithm uses the same learning rates and hyper-parameters as ES.

Table 1: Results on the MNIST digit classification task. We report the best loss and the number of steps until the training log-likelihood drops below 0.6, to observe the performance in the initial stages of learning. The values are for the best performing learning rate for each optimizer.

| Optimizer | Steps until loss $< 0.6$ | Best loss |
|---|---|---|
| ES + Adam | 433 | 0.242 |
| Ours + Adam | 182 | 0.216 |
| ES + SGD | 727 | 0.305 |
| Ours + SGD | 295 | 0.278 |

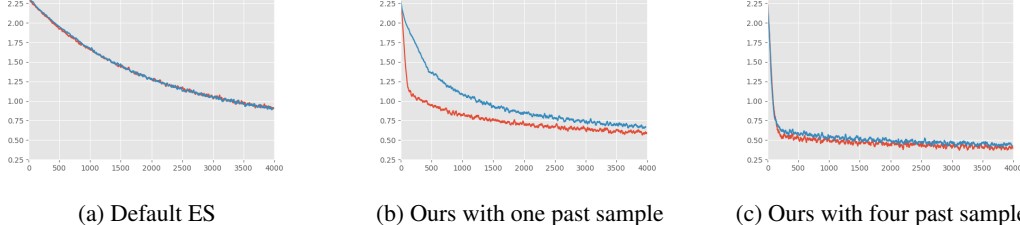

(a) Default ES     (b) Ours with one past sample     (c) Ours with four past samples

Figure 3: Using several past descent directions improves robustness to function evaluation noise. The plots show the performance on MNIST digit classification task with (blue lines) and without (red lines) function evaluation noise. The noise is created artificially by randomly permuting the fitness values of the evaluated search directions (see Equation 2) in $20\%$ of parameter updates. The x-axis represents the number of proper (i.e. non-permuted) parameter updates. (a) Standard ES does not suffer from this. (b) Function evaluation noise heavily impairs learning for our iterative gradient estimation scheme when using one past update direction as surrogate gradient. (c) Using $4$ past update directions as surrogate gradients makes our iterative gradient estimation scheme robust to function evaluation noise.

## 4.2 ROBUSTNESS TO FUNCTION EVALUATION NOISE

In practice, our iterative gradient estimation scheme may suffer from function evaluation noise because it builds up good gradient estimates over several parameter update steps. Suppose that the past update direction is a good descent direction but it performs poorly on the current batch used for evaluation due to randomness in the batch selection or network evaluation process. Then, this direction is weighted lightly when computing the new update direction, see Equation 5, and therefore the information about this direction will be discarded. We empirically show, that our approach suffers heavily from this issue when artificially injecting noise in the function evaluation process, see Figure 3b . Figure 3c shows, that this issue can be resolved by using the last $k$ update directions for our gradient estimator (see Equation 2). In this case, a good direction is only discarded, when it performs poorly in $k$ consecutive evaluation steps, which is very unlikely. We remark that the magnitude of the parameter updates naturally limits $k$, because the $k$-th last update direction is only useful if it is still correlated with the current gradient. Concretely, we found that using the last $4$ parameters updates was extremely helpful for smaller learning rates, even in the absence of noise (see Figure 3c). However, it did not offer an advantage for larger learning rates.

## 4.3 ROBOTIC RL ENVIRONMENTS

For the next set of experiments, we evaluate our algorithm on three robotics tasks of the Roboschool environment: RoboschoolInvertedPendulum-v1, RoboschoolHalfCheetah-v1 and RoboschoolAnt-v1. Our approach outperforms ES in the pendulum task, and offers a small improvement over ES in the other two tasks, see Figure 4. The improvement of our approach over standard ES is smaller on RL tasks than on the MNIST task. Therefore, we first empirically confirmed that past updated direction are also in RL correlated with the gradient. To test this, we kept track of the average difference between random perturbations and the direction given by our algorithm, after normalizing the rewards.

We found that, the direction of our algorithm had an average weight of $1.11$ versus the $0.65$ of a random direction.

RL robotics tasks bring two additional major challenges compared to the MNIST task. First, exploration is crucial to escape local optima and find new solutions, and second, the function evaluation noise is huge due to each perturbation being tested only on a single trajectory. Our proposed solution of robustness against function evaluation noise intertwines with the exploration issue. A rather small step size is necessary in order to use more past directions as surrogate gradients. However, exploration in ES is driven by large perturbation sizes and noisy optimization trajectories. We did not observe improvements when combining the approach of using several past directions with standard hyperparameter settings. We believe that an exhaustive empirical study can shed light onto the effect of our approach on exploration and may further improve the performance on RL tasks. However, running extensive experiments for complex RL environment is computationally expensive.

**Implementation details:** We use most of the hyper-parameters from the OpenAI implementation [1]. That is, two hidden layers of $256$ units each and with a $tanh$ non-linearity. Further, we use a learning rate of $0.01$, a perturbation standard deviation of $\sigma = 0.02$ and the Adam optimizer, and we also apply fitness shaping (19). For standard ES $128$ random perturbations are evaluated at each step. For our algorithm the previous gradient estimate and $126$ random perturbations are evaluated. For the Ant and Cheetah environments, we observed with this setup, that agents often get stuck in a local optima where they stay completely still, instead of running forward. As this happens for both, ES and our algorithm, we tweaked the environments in order to ensure that a true solution to the task is learned and not some some degenerate optima, we tweaked the environments in the following way. We remove the penalty for using electricity and finish the episode if the agent does not make any progress in a given amount of time. In this way, agents consistently escape the local minima. We use a $tanh$ non-linearity on the output of the network, which increased stability of training, as otherwise the output of the network would become very large without an electricity penalty.

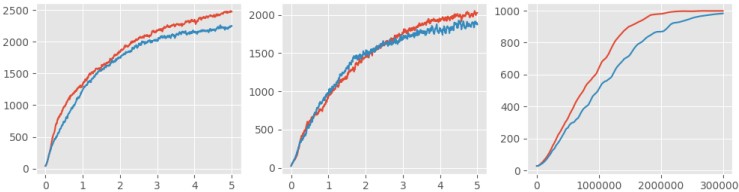

Figure 4: Performance of our algorithm (red line) and ES (blue line) on three different Roboschool tasks: Ant (left), Cheetah (center) and Pendulum (right). The plot shows the mean average reward over 9 repetitions as a function of time-steps (in thousands).

## 5 CONCLUSION

We proposed a gradient estimator that optimally incorporates surrogate gradient directions and random search directions, in the sense that it determines the direction with maximal cosine to the true gradient from the subspace of evaluated directions. Such a method has many applications as elucidated in (14). Importantly, our estimator does not require information about the quality of surrogate directions, which allows us to iteratively use past update directions as surrogate directions for our gradient estimator. We theoretically quantified the benefits of the proposed iterative gradient estimation scheme. Finally, we showed that our approach in combination with deep neural networks considerably improves the gradient estimation capabilities of ES, at no extra computational cost. The results on MNIST indicate that the speed of the Evolutionary Strategies themselves, a key part in the current Reinforcement Learning toolbox, is greatly improved. Within Reinforcement Learning an out of the box application of our algorithm yields some improvements. The smaller improvement in RL compared to MNIST is likely due to the interaction of our approach and exploration that is essential in RL. We leave it to future work to explicitly add and study appropriate exploration strategies which might unlock the true potential of our approach in RL.

---

[1] https://github.com/openai/evolution-strategies-starter

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

## A    PROOF OF THEOREMS

In this Section we prove the theorems from the main paper rigorously.

### A.1    PROOF OF PROPOSITION 1

Note that for this theorem there is no distinction between the directions $\zeta^i$ and $\epsilon^i$. For ease of notation, we denote $\zeta^1, \ldots, \zeta^k, \zeta^1, \ldots, \zeta^P$ by $\epsilon^1, \ldots, \epsilon^m$. The theorem is a simple application of the Cauchy-Schwarz inequality. Denote by $\nabla f_{\|\epsilon^i} = \sum_{i=1}^m \langle \nabla f, \hat{\epsilon}^i \rangle \hat{\epsilon}^i$ the projection of $\nabla f$ on the subspace spanned by the $\epsilon^i$s, and let $\epsilon = \sum_{i=1}^m \alpha_i \hat{\epsilon}^i$ be a vector in that subspace. Then, the Cauchy-Schwarz inequality implies

$$\langle \nabla f, \epsilon \rangle = \langle \nabla f_{\|\epsilon^i}, \epsilon \rangle \le \| \nabla f_{\|\epsilon^i} \| \|\epsilon\| . \tag{7}$$

Equality holds if and only if $\epsilon$ and $\nabla f_{\|\epsilon^i}$ have the same direction, which is equivalent to $\epsilon = \alpha g_{our}$ for some $\alpha > 0$. In particular, in this case the cosine squared between $\nabla f$ and $\epsilon$ is

$$\langle \hat{\nabla} f, \hat{\epsilon} \rangle^2 = \frac{\| \nabla f_{\|\epsilon^i} \|^2}{\| \nabla f \|^2} = \sum_{i=1}^m \langle \hat{\nabla} f, \hat{\epsilon}^i \rangle^2 \tag{8}$$

$\square$

### A.2    EXPECTATION OF COSINE SQUARED OF RANDOM VECTORS FROM THE UNIT SPHERE

We need the following proposition for the proofs of Theorem 1 and 2.

**Proposition 2.** *Let $\hat{u}$ be an $N$-dimensional unit vector, and let $\hat{\epsilon}^1, \ldots, \hat{\epsilon}^P$ be pairwise orthogonal vectors sampled uniformly from the $N$-dimensional unit sphere, that is, they are marginally uniformly distributed and conditioned to be pairwise orthogonal. Then, the expected cosine squared of $\hat{u}$ and $\epsilon = \sum_{i=1}^P \langle u, \hat{\epsilon}^i \rangle \hat{\epsilon}^i$ is*

$$\mathbb{E}[\langle \hat{u}, \hat{\epsilon} \rangle^2] = \frac{P}{N}$$

*Proof.* Note that

$$\langle \hat{u}, \hat{\epsilon} \rangle^2 = \frac{1}{\|\epsilon\|^2} \langle \hat{u}, \epsilon \rangle^2 = \frac{1}{\sum_{i=1}^P \langle \hat{u}, \hat{\epsilon}^i \rangle^2} \left( \sum_{i=1}^P \langle \hat{u}, \hat{\epsilon}^i \rangle^2 \right)^2 = \sum_{i=1}^P \langle \hat{u}, \hat{\epsilon}^i \rangle^2$$

Denote by $S$ the $N$-dimensional unit sphere. Linearity of expectation implies that

$$\mathbb{E}[\langle \hat{u}, \hat{\epsilon} \rangle^2] = \sum_{i=1}^P \mathbb{E}[\langle \hat{u}, \hat{\epsilon}^i \rangle^2]$$

$$= \frac{P}{Vol(S)} \int_S \langle \hat{u}, v \rangle^2 \, \mathrm{d}v$$

By rotational invariance of the unit sphere, we can replace $\hat{u}$ by $e_1 = (1, 0, \ldots, 0)$ and obtain

$$
\begin{aligned}
\mathbb{E}[\langle \hat{u}, \hat{\epsilon} \rangle^2] &= \frac{P}{Vol(S)} \int_S \langle e_1, v \rangle^2 \, dv \\
&= \frac{P}{Vol(S)} \int_S v_1^2 \, dv \\
&= \frac{P}{N \cdot Vol(S)} \int_S \sum_{i=1}^N v_i^2 \, dv \\
&= \frac{P}{N \cdot Vol(S)} \int_S 1 \, dv \\
&= \frac{P}{N}
\end{aligned}
$$

$\square$

### A.3 Proof of Theorem 1

In order to prove Theorem 2, use Equation 6 to compute how $X_t^2 = \langle \hat{\nabla} f, \hat{\zeta}_t \rangle^2$ depends on $X_{t-1}^2$. Then, we apply a variable transformation to $X_t$ in order to be able to apply the additive and variable drift theorems from (21), which are stated in Section B.

We can split the normalized gradient $\hat{\nabla} f = \nabla f_{\perp \hat{\zeta}_{t-1}} + \nabla f_{\| \hat{\zeta}_{t-1}}$ into an orthogonal to $\hat{\zeta}_{t-1}$ part and a parallel to $\hat{\zeta}_{t-1}$ part. It holds $\| \nabla f_{\perp \hat{\zeta}_{t-1}} \|^2 = 1 - \langle \hat{\nabla} f, \hat{\zeta}_{t-1} \rangle^2$ and $\langle \hat{\nabla} f_{\| \hat{\zeta}_{t-1}}, \hat{\epsilon}_t \rangle = 0$ since $\hat{\epsilon}_t$ is a unit vector orthogonal to $\hat{\zeta}_{t-1}$. Recall that $\hat{\nabla} f_{\perp \zeta} = \frac{\nabla f_{\perp \zeta}}{\| \nabla f_{\perp \zeta} \|}$, then

$$
\langle \hat{\nabla} f, \hat{\epsilon}_t \rangle^2 = \langle \nabla f_{\perp \hat{\zeta}_{t-1}}, \hat{\epsilon}_t \rangle^2 = (1 - \langle \hat{\nabla} f, \hat{\zeta}_{t-1} \rangle^2) \langle \hat{\nabla} f_{\perp \zeta}, \hat{\epsilon}_t \rangle^2 = (1 - X_{t-1}^2) \langle \hat{\nabla} f_{\perp \zeta}, \hat{\epsilon}_t \rangle^2 , \quad (9)
$$

and therefore by Equation 6

$$
X_t^2 = \langle \hat{\nabla} f, \hat{\zeta}_{t-1} \rangle^2 + \langle \hat{\nabla} f, \hat{\epsilon}_t \rangle^2 = X_{t-1}^2 + (1 - X_{t-1}^2) \langle \hat{\nabla} f_{\perp \zeta}, \hat{\epsilon}_t \rangle^2 . \quad (10)
$$

Define the random process $Y_t = 1 - X_t^2$. It holds

$$
Y_t = 1 - X_{t-1}^2 + (1 - X_{t-1}^2) \langle \hat{\nabla} f_{\perp \zeta}, \hat{\epsilon}_t \rangle^2 = Y_{t-1} (1 - \langle \hat{\nabla} f_{\perp \zeta}, \hat{\epsilon}_t \rangle^2) , \quad \text{and} \quad (11)
$$

$$
\mathbb{E}[Y_t | Y_{t-1} = y_{t-1}] = y_{t-1} \left( 1 - \mathbb{E}[\langle \hat{\nabla} f_{\perp \zeta}, \hat{\epsilon}_t \rangle^2] \right) = y_{t-1} \left( 1 - \frac{P}{N-1} \right) , \quad (12)
$$

where we used Proposition 2 in the $N - 1$ dimensional subspace that is orthogonal to $\zeta_{t-1}$.

In order to derive the first bound on $T$, we bound the drift of $Y_t$ for $Y_t \geq \delta$.

$$
\mathbb{E}[Y_t | Y_{t-1} = y_{t-1}, y_{t-1} \geq \delta] = y_{t-1} - y_{t-1} \frac{P}{N-1} \leq y_{t-1} - \delta \frac{P}{N-1} ,
$$

where we used Equation 12 and $y_{t-1} \geq \delta$. In order to apply Theorem 3, we define the auxiliary process $Z_t = Y_t - \delta$. Then, $T$ is the expected time that $Z_t$ hits 0. Since $\mathbb{E}[Z_t | Z_{t-1} = z_{t-1}, z_{t-1} \geq 0] \leq z_{t-1} - \delta \frac{c}{N-1}$ and $Z_0 = 1 - \delta$, Theorem 3 implies that

$$
\mathbb{E}[T] \leq \frac{1 - \delta}{\delta} \frac{N-1}{P}
$$

In order to apply Theorem 4, to show the second bound on $T$, we need to rescale $Y_t$ such that it takes values in $\{0\} \cup [1, \infty)$. Define the auxiliary process $Z_t$ by

$$
Z_t = \begin{cases} Y_t/\delta & \text{if } Y_t \geq \delta \\ 0 & \text{if } Y_t < \delta \end{cases} . \quad (13)
$$

Then, $T$ is the expected time that $Z_t$ hits 0. The process $Z_t$ satisfies

$$\mathbb{E}\left[Z_t | Z_{t-1} = z_{t-1}, z_{t-1} \geq 1\right] \leq \mathbb{E}\left[Y_t/\delta | Y_{t-1} = \delta z_{t-1}, z_{t-1} \geq 1\right] \leq z_{t-1}\left(1 - \frac{P}{N-1}\right),$$

where we used Equations 13 and 12. Since $Z_0 = 1/\delta$, Theorem 4 implies for $h(z) = z\frac{c}{N-1}$ that

$$\mathbb{E}[T] \leq \frac{N-1}{P} + \int_1^{1/\delta} \frac{N-1}{Pu} \, \mathrm{d}u = \frac{N-1}{P}(1 + \ln(1/\delta)).$$

$\square$

## A.4 PROOF OF THEOREM 2

In order to prove the theorem, we need to understand how $X_t$ depends on the value of $X_{t-1}$. It holds $X_t^2 = \langle \hat{\triangledown}f_t, \hat{\zeta}_{t-1}\rangle^2 + \langle \hat{\triangledown}f_t, \hat{\epsilon}_t\rangle^2$. As in Equation 12, we can write $\langle \hat{\triangledown}f_t, \hat{\epsilon}_t\rangle^2 = (1 - \langle \hat{\triangledown}f_t, \hat{\zeta}_{t-1}\rangle^2)\langle \hat{\triangledown}f_{\perp\hat{\zeta}_{t-1}}, \hat{\epsilon}_t\rangle^2$, and note that $\mathbb{E}[\langle \hat{\triangledown}f_{\perp\hat{\zeta}_{t-1}}, \hat{\epsilon}_t\rangle^2] = \frac{P}{N-1}$ follows by Proposition 2 and the definition of $\hat{\epsilon}_t$. This implies that

$$\mathbb{E}[X_t^2 | X_{t-1} = x_{t-1}] = \left(1 - \frac{P}{N-1}\right)\mathbb{E}[\langle \hat{\triangledown}f_t, \hat{\zeta}_{t-1}\rangle^2 | X_{t-1} = x_{t-1}] + \frac{P}{N-1} \qquad (14)$$

To understand how the $X_t$ evolves we need to analyze how $\langle \hat{\triangledown}f_t, \hat{\zeta}_{t-1}\rangle^2$ relates to $X_{t-1} = \langle \hat{\triangledown}f_{t-1}, \hat{\zeta}_{t-1}\rangle^2$. To that end, we set $\alpha_t = \langle \hat{\triangledown}f_t, \hat{\triangledown}f_{t-1}\rangle$ and write $\hat{\triangledown}f_t = \alpha_t\hat{\triangledown}f_{t-1} + \triangledown f_{\perp\triangledown f_{t-1}}$ where $\triangledown f_{\perp\triangledown f_{t-1}}$ is orthogonal to $\triangledown f_{t-1}$ and has norm $1 - \alpha_t^2$. Then,

$$\langle \hat{\triangledown}f_t, \hat{\zeta}_{t-1}\rangle^2 = \langle \alpha_t\hat{\triangledown}f_{t-1} + \triangledown f_{\perp\triangledown f_{t-1}}, \hat{\zeta}_{t-1}\rangle^2 = \left(\alpha_t\langle \hat{\triangledown}f_{t-1}, \hat{\zeta}_{t-1}\rangle + \langle \triangledown f_{\perp\triangledown f_{t-1}}, \hat{\zeta}_{t-1}\rangle\right)^2.$$

It follows that

$$\mathbb{E}[\langle \hat{\triangledown}f_t, \hat{\zeta}_{t-1}\rangle^2 | X_{t-1} = x_{t-1}] \qquad (15)$$

$$= \alpha_t^2 x_{t-1}^2 + 2\alpha_t x_{t-1}\mathbb{E}[\langle \triangledown f_{\perp\triangledown f_{t-1}}, \hat{\zeta}_{t-1}\rangle] + \mathbb{E}[\langle \triangledown f_{\perp\triangledown f_{t-1}}, \hat{\zeta}_{t-1}\rangle^2] \qquad (16)$$

$$= \alpha_t^2 x_{t-1}^2 + (1 - \alpha_t^2)(1 - x_t^2)\frac{1}{N-1}, \qquad (17)$$

where we used $\mathbb{E}[\langle \triangledown f_{\perp\triangledown f_{t-1}}, \hat{\zeta}_{t-1}\rangle] = 0$, which follows from the assumption of $\triangledown f_{\perp\triangledown f_{t-1}}$ being a random direction orthogonal to $\triangledown f_{t-1}$, and that

$$\mathbb{E}[\langle \triangledown f_{\perp\triangledown f_{t-1}}, \hat{\zeta}_{t-1}\rangle^2] = \mathbb{E}[\langle \triangledown f_{\perp\triangledown f_{t-1}}, \hat{\zeta}_{t-1\perp\triangledown f_{t-1}}\rangle^2]$$

$$= \| \triangledown f_{\perp\triangledown f_{t-1}}\|^2 \|\hat{\zeta}_{t-1\perp\triangledown f_{t-1}}\|^2 \mathbb{E}[\langle \frac{\triangledown f_{\perp\triangledown f_{t-1}}}{\| \triangledown f_{\perp\triangledown f_{t-1}}\|}, \frac{\hat{\zeta}_{t-1\perp\triangledown f_{t-1}}}{\|\hat{\zeta}_{t-1\perp\triangledown f_{t-1}}\|}\rangle^2]$$

$$= (1 - \alpha_t^2)(1 - x_t^2)\frac{1}{N-1},$$

which follows from $\| \triangledown f_{\perp\triangledown f_{t-1}}\|^2 = 1 - \alpha^2$, $\|\hat{\zeta}_{t-1\perp\triangledown f_{t-1}}\|^2 = 1 - x_t^2$ and Proposition 2 for $P = 1$ using that $\triangledown f_{\perp\triangledown f_{t-1}}$ is a random direction orthogonal to $\triangledown f_{t-1}$. Then, plugging Equation (17) into (14), implies the theorem. $\square$

## B DRIFT THEOREMS

For the proof of Theorem 1, we use two drift theorems from (21), which we restate for completeness.

**Theorem 3** (Additive Drift, Theorem 1 from (21)). *Let $(X_t)_{t\in\mathbb{N}_0}$ be a Markov chain with state space $S \subset [0, \infty)$ and assume $X_0 = n$. Let $T$ be the earliest point in time $t \geq 0$ such that $X_t = 0$. If there exists $c > 0$ such that for all $x \in S$, $x > 0$ and for all $t \geq 0$ we have*

$$\mathbb{E}[X_{t+1} | X_t = x] \leq x - c.$$

*Then,*

$$\mathbb{E}[T] \leq \frac{n}{c}.$$

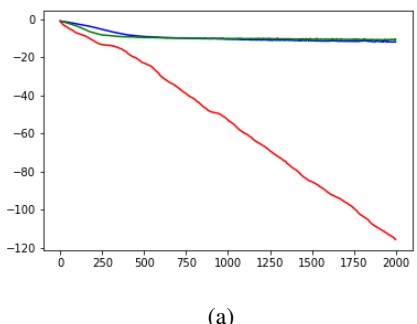 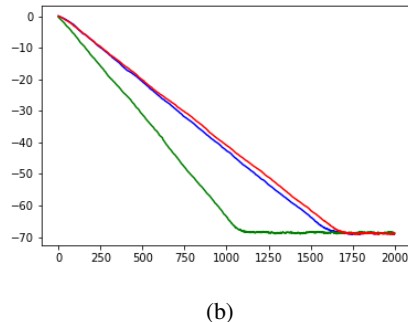

          (a)                                                   (b)

Figure 5: Performance when optimizing a quadratic function. Performances of our approach (green), standard ES (blue) and SNES (red). (a) On small dimensional functions ($N = 32$ and 16 sampled directions per time step) SNES outperforms our approach and standard ES. Our approach and ES do not improve further because they do not adapt their stepsize. (b) For high dimensional functions ($N = 10000$ and 16 sampled directions per time step), our approach outperforms SNES and standard ES.

**Theorem 4.** *Variable Drift, Theorem* 4 *from (21)] Let* $(X_t)_{t \in \mathbb{N}}$ *be a Markov chain with state space* $S \subset \{0\} \cup [1, \infty)$ *and with* $X_0 = n$. *Let* $T$ *be the earliest point in time* $t \geq 0$ *such that* $X_t = 0$. *Suppose furthermore that there is a positive, increasing function* $h : [1, \infty) \to \mathbb{R}_{>0}$ *such that for all* $x \in S$, $x > 0$ *we have for all* $t \geq 0$

$$\mathbb{E}[X_{t+1} | X_t = x] \leq x - h(x) .$$

*Then,*

$$\mathbb{E}[T] \leq \frac{1}{h(1)} + \int_1^n \frac{1}{h(u)} \, \mathrm{du} .$$

## C   ADDITIONAL EXPERIMENTS

This section presents some additional experiments. We will incorporate a polished version of this section into the experiments section of the main paper, for the camera ready version of the paper.

### C.1   COMPARISON OF OUR APPROACH TO SEPARABLE NES AND GUIDED ES

We consider the toy task of optimizing a quadratic function $f(x) = \|Ax\|_2^2$ as considered in (14). Figure 5 compares our approach to a diagonal approximation of CMA-ES, that is, seperable natural ES (SNES) from (19), that has the same runtime and memory complexity as our approach. While SNES works verv well for small diminsional parameter space ($N = 32$ , see Figrue 5a ), it is clearly outperformed by our approach for high dimensional paramater spaces ( $N = 10000$, see Figure 5b).

Further, we compare our one step gradient estimator $g_\perp$ to the guided-ES gradient estimator proposed in (14). Again the goal is to optimize a quadratic function $f(x) = \|Ax\|_2^2$. Both estimators receive a surrogate gradient $\zeta$ for the optimization. The surrogate gradient is created as in (14): A normalized random bias $b$ (drawn once at the beginning of optimization) and a normalized noise direction $n$ (resampled at every iteration) are added to the true gradient. that is, $\zeta(x) = \bigtriangledown f(x) + (b+n)\|\bigtriangledown f(x)\|_2$ Figure 6 shows that our approach outperforms guided-ES no matter how the parameter $\alpha$ controlling the bias variance trade-off of guided-ES is set.

**Implementation details:** For Figure 5b, $x$ has dimension $N = 10000$ and is initalized by sampling $x_0 \sim \mathcal{N}(0, I)$. $A$ is a matrix of dimension $100 \times 10000$, and its entries are samples from a $\mathcal{N}(0, 0.001)$ distribution, where the variance is chosen such that the initial loss is approx 1. For Figure 5a, $x$ has dimension $N = 16$ and $A$ has dimension $16 \times 16$. The hyperparamters for our approach and standard ES are the learning rate $\beta$ and perturbation size $\sigma$. The hyperparamters for SNES are learning rate

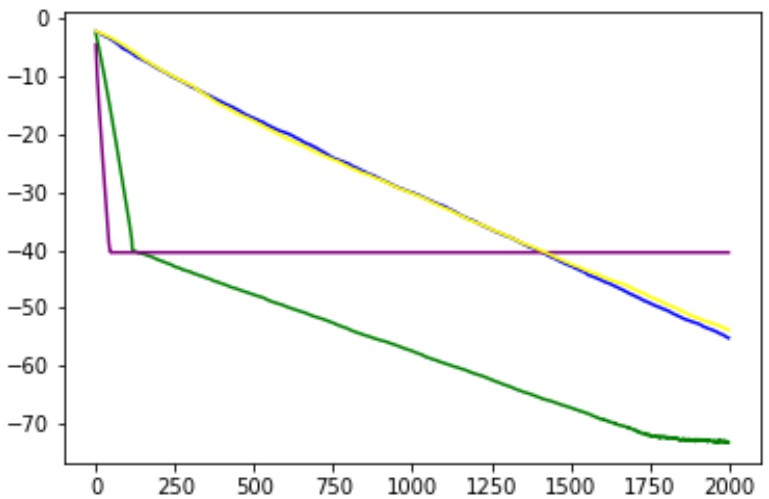

Figure 6: Optimizing a quadratic function with biased surrogate gradients. Performances of our approach (green line), SGD using the surrugate gradient for the parameter update (purple line), ES (blue line), guided-ES (yellow line) are plotted. Our approach (green line) improves sharply, making use of the biased gradient, until it becomes useless (crossing with purple line). For guided-ES, we observed a binary behaviour when optimizing for the parameter that controls the bias-variance trade-off analysed in (14). Either it follows the surrogate gradient very closely or it behaves similarly to ES. This observation is consistent with the analysis conducted in (14). The optimal value that we found for this parameter is very close to 1 and results in a performance very close to ES (yellow and blue lines).

$\beta$, learning rate $\beta_\sigma$ for the step size adaptation and perturbation size initialization. We performed a hyperparameter search for all these hyperparameters and plotted the best performance.

