# OpenReview forum: "Improving Gradient Estimation in Evolutionary Strategies With Past Descent Directions"
_ICLR.cc/2020/Conference — Reject_

### Official Review · AnonReviewer1 · 2019-10-23
**Official Blind Review #1**

**Rating:** 3

**Review:**

This paper addresses the issue of noisy gradient estimation in a type of evolution strategies popularized by the open AI's reinforcement learning paper. It is a follow-up paper of reference [14], and try to analyze the optimality of the gradient estimation. The goal of the paper is well stated and well motivated. The paper itself is well-organized. However, the novelty of this work is not sufficiently high and its usefulness is questionable.

Theorem 1 is trivial under the assumption stated above the theorem---the numerical approximation of the directional derivative admits the true directional derivative. In other words, the assumption is too strong to claim the goodness of the proposed scheme.

Let's just sample k+P random normal vectors and orthogonalize and normalize them. Let them denoted by the same symbols hat zeta and hat epsilon. The theorem statements holds for this case. Therefore I failed to understand the essential claim of Theorem 1.

About Theorem 2 and 3, again, the assumption that he numerical approximation of the directional derivative matches the true directional derivative is too strong to make the claim relevant. Moreover, the effect of P somehow disappear from the analysis.

All the analysis is done assuming the above mentioned strong assumption. However, in one of the experiments and the existing works, ranking based fitness shaping has been applied to make the algorithm robust. This replaces the function value differences in the gradient estimator with some predefined values depending on the ranking of f-values of each trial vector. This definitely violates the assumption, and it may result in some vector far away from the true gradient, yet the algorithm still works well. Therefore, the hypothesis underlying in this paper---better estimation of the gradient will lead to a better performance---may not be true. At least the numerical experiments provided in this paper do justify this hypothesis.

The numerical experiments have been conducted to compare the proposed algorithm with the baseline ES algorithm. In a sense it is reasonable to evaluate the effect of the proposed modification in the baseline ES. However, since the baseline ES algorithm is not really efficient on tasks such as the one conducted in Figure 2, the usefulness of the proposed approach is not tested. At least one should compare with the "canonical" ES, where the learning rate fixed and sigma is adapted. See https://arxiv.org/pdf/1802.08842.pdf.

Use of the search history proposed in this paper is not really new in ES community. A sort of momentum terms appears in the standard CMA-ES [18] even in two parts of the algorithm and its effectiveness is well-studied empirically. This paper addresses the theoretical aspect of the momentum and this may be new. However, as mentioned above, the assumption is too strong to describe the reality.

"Linear time approximations of CMA-ES like diagonal approximations of the covariance matrix (19) often do not work well." Please specify in what sense the linear time version of the CMA-ES do not work well and provide the evidence (references).

**Experience Assessment:**

I have published in this field for several years.

**Review Assessment: Checking Correctness Of Derivations And Theory:**

I assessed the sensibility of the derivations and theory.

**Review Assessment: Checking Correctness Of Experiments:**

I carefully checked the experiments.

**Review Assessment: Thoroughness In Paper Reading:**

I read the paper at least twice and used my best judgement in assessing the paper.

---

> ### Author Response · Authors · 2019-11-15
> **Response to Reviewer 1**
>
> We agree that Theorem one follows very easily from that assumption. Therefore, we renamed it to Proposition 1.
>
> We shortly discuss how reasonable the assumption is that the numerical approximation is equal to the true directional derivative. The assumption can be violated because of two reasons: 1) third order and higher order terms make the approximation imprecise, and 2) function evaluation noise (due to random RL environments or stochastic initialization) make the numerical approximation imprecise.
> 1) We agree that this is an issue. However since the success of momentum based approaches established that higher order terms are often ‘well behaved’ in Deep Learning, we believe that our assumption is reasonable.
> 2) Large function evaluation noise is often an issue (at least in RL). However, in such a scenario the statement of Proposition 1 holds for the expectation of our gradient estimator. That is, E[g_our] is equal to the direction in the aforementioned subspace that is most aligned with the gradient.)
>
> We included the effect of P into our analysis.
>
> We agree that using ES with rank based fitness shaping is not the ideal choice to show that better gradient estimation leads to better performance. There seems to be a bias - variance trade-off here, that favours reducing the variance by fitness shaping. Still, using past descent directions seems to improve gradient estimation quality also in this setting and leads to improved performance. Further, the fitness shaping method is widely used in practise. Therefore, we think that showing that past update directions improve performance  of that method (Figure 4) has the highest practical impact.
>
> In the new plots we compared our approach to diagonal approximations of CMA-ES, which are more general than the canonical ES, since the sigma is adapted for every parameter separately.
>
> We clarified the comment about diagonal CMA-ES in the paper and in the general reply to the reviewers above.

---

### Official Review · AnonReviewer3 · 2019-10-23
**Official Blind Review #3**

**Rating:** 3

**Review:**

This paper is about improving the quality of surrogate gradients. Their proposal in guaranteed to find a descent direction. In addition to two results, the authors also provide experimental results.

This paper is presenting a research on a recent topic. Using random search methods or evolutionary strategies in machine learning problems is attracting quite an interest in the last years. However, this particular paper misses several important points and hence, lacks sufficient contribution.

Here are my major comments:

- The technical results are somewhat superficial: The first one is with the big assumption of linearity. This assumption does not hold almost for all problems where random search strategies would be of use. The second result, on the other hand, assumes orthogonality, which almost surely never happens. I must add that the authors also acknowledge the severity of these assumptions.

- It is important to note that the theory here assumes that the gradient does exist but cannot be computed or too expensive to compute.

- I would have expected an experimental study that would properly support the proposed approach. Such a study would have shed light on the computation time and efficiency. The authors solve MNIST problem, which can be quite efficiently solved with a variant of (accelerated) gradient method. Solving it with ES and then improving the result with the proposed approach is not satisfactory. Reinforcement learning experiments could have been noteworthy but the authors have solved quite small problems and did not compare their results extensively against contender approaches like augmented random search. It would have been also nice to see the computation times on large problems since the extensive computation time is a big obstacle for training in reinforcement learning.


**Experience Assessment:**

I have read many papers in this area.

**Review Assessment: Checking Correctness Of Derivations And Theory:**

I assessed the sensibility of the derivations and theory.

**Review Assessment: Checking Correctness Of Experiments:**

I assessed the sensibility of the experiments.

**Review Assessment: Thoroughness In Paper Reading:**

I made a quick assessment of this paper.

---

> ### Author Response · Authors · 2019-11-15
> **Response to Reviewer 3**
>
> 1) We agree that linearity is a big assumption. However, it is desirable that the gradient estimate converges to the true gradient in the linear case (Note that for example standard ES, diagonal CMA-ES do not do so).  Furthermore, we think it is interesting how fast it does so. We added to a few lines of explanations why we consider this case and how the speed of convergence relates to the one of orthogonal sampling.
> For the second result, we want to clarify that we do not assume orthogonality, since any vector can be partitioned into a parallel and orthogonal part. The assumption is that the direction of the orthogonal part is a random orthogonal direction. This implies that the direction of rotation of the gradient when doing a parameter update step is uncorrelated with the previous gradient estimate zeta. Which is what is actually required in the proof. Though this is of course an assumption that is not true in general, we do not see any reason why the gradient should rotated away from zeta (which would be the scenario in which the gradient estimation is worse that predicted by our theory).
>
> 2) Yes, we assume differentiability for the analysis. We stated this in the paper (before Proposition 1).
>
> 3)
> -The MNIST example serves as a proof of concept and to evaluate the quality gradient approximation, which can be only directly evaluated if the true gradients can be computed.
> - Thanks for the suggestion of using augmented random search as baseline. Time did not suffice, but we consider adding it to the plots.
> - The computation time of our approach is exactly the same as the one of ES on any problem size.

---

### Official Review · AnonReviewer2 · 2019-10-24
**Official Blind Review #2**

**Rating:** 6

**Review:**

This paper provides a new type of gradient estimator that combines an Evolutionary Strategies (ES) style estimate (using function evaluations at perturbed parameters) along with surrogate gradient estimates (gradient estimates that may be biased and/or high variance). The estimator involves computing antithetic ES estimates in two subspaces: along the set of (normalized) surrogate gradients, and along a set of randomly chosen vectors in the orthogonal complement of the span of the surrogate gradients. The paper provides a proof of the optimality of the estimate, that is, the proposed gradient estimate maximizes the cosine of the angle with the true gradient over the vectors in the subspace defined by the set of surrogate gradients and sampled directions. The paper proposes an additional mechanism for generating surrogate gradients by simply using previous gradient estimates as surrogate gradients, and derives a convergence rate for when this iterative estimator will approximate a fixed, true gradient (e.g. for linear functions). Finally, the paper applies the estimate to two tasks: MNIST classification and robotic control via reinforcement learning, demonstrating improvements on both compared to standard ES.

I think this is a nice contribution, and I enjoyed reading this paper, with one major caveat regarding some of the experiments. The paper is clearly written.

Major concerns:
- The paper is missing critical comparisons to existing work. In particular, the paper cites Ref. 14 as another method for using surrogate gradients in optimization. For both examples (MNIST and RL), it is crucial to add the algorithm from that paper as a baseline.
- In addition, it would also be nice to see one of the diagonal approximations of CMA-ES as a baseline.

Other questions/comments:
- For the MNIST example, you mentioned that the function is deterministic--how many examples are used for each function/gradient evaluation (the full dataset, or some fixed subset)?
- It would be nice to see how the performance gap between the proposed estimate and ES varies with the number of parameters (size of the network).
- It would be nice to compare the orthogonal epsilon to the N(0,I) epsilon case. As mentioned in the paper, the N(0,I) will be nearly orthogonal to the surrogate gradients in high dimensions. For a practical problem (e.g. MNIST), is the orthogonalization strictly necessary?

Minor comments/typos:
- Fig 2: Add label for the x- and y-axes, and a legend.
- Use a more semantically meaningful subscript than `our` for the proposed gradient estimate. Perhaps `orth`, since you utilize orthogonal subspaces?
- Typo in eq. (6) (issue with the subscript on f)


**Experience Assessment:**

I have published one or two papers in this area.

**Review Assessment: Checking Correctness Of Derivations And Theory:**

I assessed the sensibility of the derivations and theory.

**Review Assessment: Checking Correctness Of Experiments:**

I carefully checked the experiments.

**Review Assessment: Thoroughness In Paper Reading:**

I read the paper thoroughly.

---

> ### Author Response · Authors · 2019-11-15
> **Answer to reviewer 2**
>
> -We added comparisons to Guided-ES (Ref. 14) and diagonal-CMA-ES  (seperable NES) for a high-dimensional quadratic function, to adress the major concerns.
>
> -All samples used the exact same mini-batch (resampled after each ES update). The batch has size 100.
> -We agree that it is interesting to investigate how this performance gap depends on the number of parameters, but we did not have enough time to create this plot.
> -We observed that the orthogonal epsilon and the N(0,I) epsilon cases have the same performance. The plots are produced using N(0,I) epsilons.
>
> -We will add axis labels for the camera ready version.
> -We use the ||\epsilon subscript in that Equation to denote the part of the vector that is parallel to \epsilon

---

### Author Response · Authors · 2019-11-15
**General Response to the Reviewers**

We thank the reviewer for their valuable comments that helped us improving our manuscript.

Let us first highlight the main changes of the manuscript:
-We explained more clearly the purpose of the theorems (first paragraph of Section 3.3).
-We included the number of samples P into the analysis. This allows us to validate theoretical predictions empirically, see second paragraph below.
-We demonstrate that our approach clearly outperforms other contender approaches like diagonal CMA-ES and Guided-ES at the example of optimizing a high-dimensional quadratic function as done in (14). These plots are available in Appendix.C now. These will be polished and moved to the main text for the final version.

We want to emphasize that the experiments on the quadratic function and MNIST serve as a proof of concept to demonstrate the improved gradient estimation quality, while the improved performance over standard ES for the tested RL enviromnents reveals the practical impact that our approach may have, as standard ES is a commonly used method. Especially, this improvement comes without computational extracost and without increased implementation complexity.

For theorem 2 we assume that the true gradient is modified by independent orthogonal noise. While this may seem like a big assumption at first, there is no particular reason to believe the gradient will change in a way that is adversarial to our algorithm. To test this, we have tracked the change of the true gradient over training (alpha) and computed the expected improvement in gradient estimation, under our assumptions and with the observed alphas. This gave us an expected improvement of 9.2e-7 over taking random orthogonal samples, which would be the square root of P/(N-1) (P=number of samples, N=dimensionality of the problem).
However, we found that taking random samples in ES gave a worse than expected cosine, by -4.1e-7, which could be explained because of the influence of higher order terms  or because our samples are independently sampled and not pairwise orthogonal. Taking this into account, we computed our observed improvement over the expectation to be 5.2e-7, which once we subtract the loss we observed (from higher order terms or non-orthogonality) we get a value of 9.1e-7 which is extremely close to our theoretical predictions.

We believe these experiments shows that are assumptions are very reasonable and our theory closely models the observed behaviour. We will add this experiments and explanation to the final version of the paper.

Further, we want to emphasize that fast convergence to the true gradient for linear function as shown in our theorem is a desirable property (as it increases the quality of the gradient estimation in the case of constant or very slowly changing gradients), that is not satisfied by many other approaches like standard ES, canonical ES, diagonal approximations of CMA-ES. This is clear for standard ES. But also for diagonal approximations of CMA-ES, gradient estimation is not improved if the gradient is not aligned with the coordinate axis, e.g. if the gradient is (1,1,...,1) then all coordinates affect the loss equally and the sampling scheme will not differ from the one of standard ES as all sigmas will be the same. Therefore, in terms of gradient estimation our algorithm clearly outperforms all these approaches.

We address the further concerns of the reviewers in seperate answers.

---

### Decision · Program_Chairs · 2019-12-19

**Decision:**

Reject

**Comment:**

The authors propose a novel approach to using surrogate gradient information in ES. Unlike previous approaches, their method always finds a descent direction that is better than the surrogate gradient. This allows them to use previous gradient estimates as the surrogate gradient. They prove results for the linear case and under simplifying assumptions that it extends beyond the linear case. Finally, they evaluate on MNIST and RL tasks and show improvements over ES.

After the revisions, reviewers were concerned about:
* The strong (and potentially unrealistic) assumptions for the theorems. They felt that these assumptions trivialized the theorems.
* Limited experiments demonstrating advantages in situations where other more effective methods could be used. The performance on the RL tasks shows small gains compared to a vanilla ES approach. Thus, the usefulness of the approach is not clearly demonstrated.

I think that the paper has the potential to be a strong submission if the authors can extend their experiments to more complex problems and demonstrate gains. At this time however, I recommend rejection.